# Surgical, Histopathological, and Quality of Life Outcomes Following Neoadjuvant Chemotherapy and Pancreatectomy for Borderline Resectable and Locally Advanced Pancreatic Cancer [note 1]

**DOI:** 10.3390/cancers17152505

**Published:** 2025-07-29

**Authors:** Ingvild Farnes, Caroline S. Verbeke, Dyre Kleive, Anne Waage, Tore Tholfsen, Milada Hagen, Bjarte Fosby, Pål-Dag Line, Knut Jørgen Labori

**Affiliations:** 1Department of Hepato-Pancreato-Biliary Surgery, Oslo University Hospital, Rikshospitalet, 0372 Oslo, Norway; infarn@ous-hf.no (I.F.); dyrkle@ous-hf.no (D.K.); uxawaa@ous-hf.no (A.W.); tortho@ous-hf.no (T.T.); 2Institute of Clinical Medicine, University of Oslo, 0318 Oslo, Norway; c.s.verbeke@medisin.uio.no (C.S.V.); p.d.line@medisin.uio.no (P.-D.L.); 3Department of Pathology, Oslo University Hospital, Rikshospitalet, 0372 Oslo, Norway; 4Department of Health Science and Biostatistics, Oslo Metropolitan University, 0130 Oslo, Norway; milasm@oslomet.no; 5Department of Transplantation Medicine, Oslo University Hospital, 0372 Oslo, Norway; bfosby@ous-hf.no

**Keywords:** pancreatic cancer, borderline resectable, locally advanced, neoadjuvant therapy, surgical complications, quality of life, histopathology, overall survival

## Abstract

This prospective study examines outcomes for patients with borderline resectable and locally advanced pancreatic cancer who received chemotherapy before surgery. The aim was to see how this treatment combination affects survival and quality of life. We analyzed data from 54 patients who underwent neoadjuvant chemotherapy and pancreatectomy. Most patients had complex surgery, including vascular resections in 59.3%, and 25.9% faced major complications. However, the results show that survival rates in these patients were similar to those seen in less advanced cases with primary resectable pancreatic cancer. Importantly, quality of life improved within three months after surgery. Only a high tumor regression grade after treatment was linked to worse survival. These findings support the use of this treatment approach, showing that it offers improved survival and recovery of quality of life in pancreatic cancer patients with major vascular involvement.

## 1. Introduction

Pancreatic cancer is one of the most lethal cancers, with the 5-year overall survival rate being less than 5% for all stages combined. By 2030, pancreatic cancer will be the second leading cause of cancer-related death [1]. Approximately 35% of the patients have borderline resectable (BRPC) or locally advanced pancreatic cancer (LAPC) at the time of diagnosis [2]. Following primary chemotherapy, resection is the only potentially curative treatment. In recent years, the introduction of more effective multimodal chemotherapy regimens has resulted in increased overall survival (OS) and resection rates for these patients [2]. Parallel to these advancements in systemic therapy, significant progress has also been achieved in surgical techniques, including complex vascular resections and reconstructions, thereby expanding the boundaries of what is considered resectable disease [2,3,4].

A recent population-based study from Norway by our group provided real-world evidence of these evolving trends, reporting a 47% resection rate in BRPC and 13% in LAPC [5]. In the multivariate analysis of baseline factors associated with resection, LAPC, carbohydrate antigen 19-9 (CA19-9) > 500, and treatment with gemcitabine were negative predictors of eventual resection in the 188 patients initiating primary chemotherapy [5]. However, selection of patients for resection after primary chemotherapy remains challenging [6,7]. The overall dismal prognosis of pancreatic cancer and the complex vascular resections often needed in BRPC/LAPC require optimal preoperative evaluation and clinical decision-making to achieve the best surgical and histopathological outcomes for improved survival. Moreover, current guidelines recommend that, in addition to delivering favorable oncological outcomes, surgery should be performed efficiently with the goal of improving patients’ health-related quality of life (QoL) [8]. Therefore, in addition to clinical and oncological considerations, it is equally important to address the patients’ physical and psychosocial experience, and for this reason, incorporating assessments of health-related QoL measurements into the clinical decision-making process is important.

The present study is a subgroup analysis derived from the aforementioned Norwegian population-based cohort, focusing specifically on patients with BRPC and LAPC who underwent resection after neoadjuvant chemotherapy [5]. The primary aim of the study was to evaluate the clinical course of these patients, with specific emphasis on perioperative and postoperative outcomes, detailed histopathological findings, OS, and QoL.

## 2. Materials and Methods

### 2.1. Study Population, Study Design, and Definitions

From January 2018 to December 2020, all patients with BRPC/LAPC in the South-Eastern Norway Regional Health Authority referred to the multidisciplinary team (MDT) at Oslo University Hospital (OUH) were offered participation in the NORPACT-2 study and included following written informed consent. The study protocol was approved by the Regional Ethics Committee (REK Nord 2017/1382) and conducted in accordance with the STROBE guidelines [9].

The protocol for overall treatment followed the national Norwegian guidelines, as previously described [5,10]. BRPC/LAPC and vascular involvements were classified according to NCCN criteria version 2, 2017, by an experienced team of abdominal radiologists and pancreatic surgeons [11]. Endoscopic ultrasound-guided fine-needle aspiration cytology/biopsy was required to confirm pancreatic cancer [12]. Performance status (Eastern Cooperative Oncology Group (ECOG)) and the Charlson comorbidity index were recorded at the time of diagnosis. Patients were followed up until date of death or until 12 December 2023. Radiological response evaluation was defined in accordance with the Response Evaluation Criteria in Solid Tumors (RECIST) [13]. OS was calculated from time of diagnosis and time of surgery to time of death or date of last follow-up. Recurrence was defined as radiological evidence of locoregional (soft tissue regionally or in the former surgical bed) or distant metastasis. Recurrence-free survival (RFS) and post-recurrence survival were calculated from time of surgery and time of cancer-related recurrence, respectively, to time of death or date of last follow-up.

### 2.2. Postoperative Complications, Histopathology Examination, and Quality of Life

Postoperative course, 90-day postoperative complications, and readmission rates were recorded from Oslo University Hospital records and from medical records from the patient’s local hospital. Clavien–Dindo classification, the Charlson comorbidity index (https://www.mdcalc.com/charlson-comorbidity-index-cci, accessed on 15 May 2025), and adverse events were recorded for all patients [14,15]. Post-pancreatectomy hemorrhage (PPH), delayed gastric emptying (DGE), and postoperative pancreatic fistula (POPF) were registered according to the International Study Group of Pancreatic Surgery definitions [16,17,18].

All surgical specimens were evaluated by one experienced pathologist (CSV). Histopathological evaluation was undertaken according to a fully standardized protocol. Tumor size was evaluated based on combined macro- and microscopical measurement in three dimensions [11]. Local tumor extent (ypT-stage) and regional lymph node metastasis were evaluated according to UICC TNM 8th edition [19]. The relationship of the tumor and specimen surfaces and margins was examined according to the dataset provided by The Royal College of Pathologists [20]. In brief, following color-coded inking of the pancreatic surfaces (anterior, posterior, towards the superior mesenteric artery and vein (“SMV groove”), respectively), the formalin-fixed specimen was dissected by axial slicing [21]. The pancreas and any other abnormal-looking tissues were completely embedded, with all tissue blocks including the overlying tissue surface to allow detailed assessment of the resection margins and surfaces. The presence of invasive carcinoma within 1 mm to a margin and 0 mm to the anterior pancreatic surface was reported as R1, according to (inter-)national guidelines [22]. The histopathological tumor response to neoadjuvant chemotherapy was assessed according to the College of American Pathologists (CAP) tumor regression grading system, which distinguishes between the following 4 grades: no viable cancer cells (CAP0), single cells or rare small groups of cancer cells (CAP1), residual cancer with evident tumor regression but more than single cells or rare small groups of cancer cells (CAP2), and extensive residual cancer without evident tumor regression (CAP3) [23].

The European Organization for Research and Treatment Center Quality of Life Questionnaire C30 (EORTC QLQ-C30) was used to assess patient QoL. The EORTC QLQ-C30 outcomes were transformed into a continuous scale from 0 to 100 [24]. QoL questionnaires were collected at baseline, before surgery, and at 6 weeks, 3, 6, and 12 months postoperatively. EORTC QLQ-C30 comprises 30 questions constituting functioning and symptom scales. The two final questions measure overall health and global QoL, forming the global QoL score. Responses are given on an ordinal scale encompassing 1–4 categorical scales ranging from not at all, a little, quite a bit, and very much. The two modified visual analog scales assessing global health and QoL range from 1 to 7, from very poor to excellent. Higher scores on the symptom scales from the questionnaires indicate more severe symptoms, while higher scores on the functional scales indicate better functioning [24].

### 2.3. Statistics

Statistical analyses, QoL analyses, and survival curves were performed using SPSS version 29.0.0.0. (241). Cox regression uni- and multivariable analyses were performed using SPSS version 29.0.0.0. (241) and Stata 18 v. 14 February 2024 (x21.18.0.132). Continuous variables are described as medians with interquartile ranges (IQRs). Categorical variables are expressed as counts with %. OS and RFS were estimated by the Kaplan–Meier method with a set 95% confidence interval. Univariate and multivariate (for relevant variables) associations for OS were assessed using a Cox regression model. Variables that were statistically significant in univariate analyses were included in a multivariable model. We have generalized mixed models (GLMs) for repeated measures to assess possible changes over time for QoL variables. We used identity link function as the outcome was continuous. GLMs do not require complete data, so we could use all available data despite dropouts and missing data. Moreover, we used an unstructured covariance matrix so as not to impose any structure on our data. Statistical significance was set at *p* < 0.05.

## 3. Results

### 3.1. Overall Cohort Characteristics

Of a total of 188 patients with BRPC or LAPC initiating primary chemotherapy, 67 patients with histology-proven BRPC/LAPC underwent neoadjuvant chemotherapy and subsequent surgical exploration (Figure 1). Overall, 54 patients underwent a successful surgical resection (Figure 1). Thirteen patients were explored and had a non-therapeutic laparotomy due to locally advanced disease (n = 7), liver metastasis (n = 2), or peritoneal metastasis (n = 4). Eight (61.5%) of these patients continued with palliative chemotherapy (mFOLFIRINOX n = 3, gemcitabine/nab-paclitaxel n = 3, gemcitabine n = 2). The remaining five patients received best supportive care. Baseline characteristics for the 54 patients undergoing resection are presented in Table 1. Forty-three (79.6%) patients had BRPC and eleven (20.4%) had LAPC. Forty-seven (87%) patients had tumors in the pancreatic head/uncinate process. On primary CT, 45 (83.3%) patients had portal–superior mesenteric vein involvement, 34 (63.1%) patients had arterial involvement, and 19 (35.2%) patients had both arterial and venous involvement.

### 3.2. Neoadjuvant Chemotherapy and Response Evaluation

(m)FOLFIRINOX was used in 36 (66.7%) patients, and gemcitabine/nab-paclitaxel was used in 13 (24%) (Table 1). Only three (5.6%) patients underwent a chemotherapeutic switch. The median number of cycles was four (IQR 2–6). CA19-9 normalized during neoadjuvant chemotherapy in 10 (18.5%) patients and decreased > 50% in 22 (40.7%). Eight (14.8%) patients had partial RECIST response at restaging.

### 3.3. Surgical Procedure, Peri- and Postoperative Outcome, and Adjuvant Chemotherapy

Details on the surgical procedures and related postoperative complications and outcomes are listed in Table 2. Forty-six (85.2%) patients underwent pancreatoduodenectomy, five (9.3%) patients underwent total pancreatectomy with splenectomy, and three (5.6%) patients underwent open distal pancreatectomy with splenectomy. Thirty-two (59.3%) patients underwent vascular resection. Six (11.1%) had a combined venous and arterial resection. The median operative time and intraoperative blood loss were 444 min and 400 mL, respectively. Fourteen (25.9%) patients experienced major complications (Clavien Dindo ≥ 3) within 90 days postoperatively (Appendix A). The most common minor complication was delayed gastric emptying in eight (14.8%) patients. One (2%) patient developed a clinically relevant POPF. Reoperations occurred in three (5.6%) patients. There was no 90-day mortality.

### 3.4. Pathology Assessment and Histopathological Outcomes

Histopathological findings are detailed in Table 3. The majority of the surgical specimens were staged as ypT2 (n = 34 (63%) and ypN1 (n = 25 (46.3%)). The median tumor size was 33 mm (IQR 27–39). The median lymph node yield was 21 (range 17–29), and the median lymph node ratio was 0.08 (range 0.05–0.18). The majority of specimens showed both perineural and lymphatic invasion (n = 50 (92.6%) and n = 37 (68.5%)), whereas microvascular invasion was less common (n = 21 (38.9%)). One (1.9%) patient had complete tumor regression, while in the remaining patients, tumor regression was graded as CAP 1 (n = 3, 5.7%), CAP 2 (n = 26, 48.1%), and CAP 3 (n = 23, 42.6%). There were seven (13%) R0 resections, forty-six (85.2%) R1 resections, and one (1.9%) R2 resection (Table 3).

The relationship of the tumor to the specimen margin sites was assessed (Appendix A). One margin was involved in eighteen (33.3%) cases, two margins were involved in fourteen (25.9%) cases, three margins were involved in five (9.3%) cases, and four margins were involved in three (5.6%) cases. The most frequently affected surface was the superior mesenteric vein (SMV) groove (in 25 specimens (49%)), followed by the superior mesenteric artery (SMA) surface (in 19 specimens, 37.3%). In all specimens, the proximal gastric/duodenal and bile duct resection margins were clear.

### 3.5. Survival, Recurrence, and Prognostic Factors

Median OS for the 54 patients who underwent a successful resection was 31 (CI 24.7–37.3) months (Figure 2). In BRPC and LAPC patients, OS was 29 (CI 18.7–39.3) and 34 (CI 20.1–48) months, respectively (*p* = 0.757). Median OS from time of surgery was 26 (CI 16.3–35.7) months (Appendix A), and it was 25 (CI 14.7–35.3) months for BRPC and 29 (CI 10.3–48) months for LAPC (*p* = 0.816).

Median RFS was 12 (CI 7.9–16.1) months (Appendix A), and it was 15 (CI 11–199 months for BRPC and 6 (CI 1.7–10.3) months for LAPC (*p* = 0.144). There was a significant difference in OS in patients ≤ 70 years (43 months (CI 29.2–56.8)) versus patients >70 years (23 months (CI 15.2–30.8)) (*p* = 0.008). Median OS was 13.1 (CI 1.5–24.7) months for the 13 patients who underwent a non-therapeutic laparotomy.

Thirty-nine (72.2%) patients experienced recurrence (pancreatoduodenectomy n = 34, distal pancreatectomy n = 1, total pancreatectomy n = 4) (Appendix A). Site of first recurrence was locoregional recurrence only in eleven patients (28.2%), combined locoregional recurrence and distant metastases in nine (23.1%), and distant metastases at single or multiple sites in nineteen (48.7%). Liver-only recurrence was registered in six patients and lung-only in three patients.

In univariate analyses, age > 70 (HR 2.29, *p* = 0.011), CA19-9 > 150 at response evaluation (HR 2.61, *p* = 0.017), initiating gemcitabine-based chemotherapy (HR 2.18, *p* = 0.017), CAP grade (CAP 3 versus CAP 0–2, HR 3.36, *p* < 0.001), N2 status (HR 3.06, *p* = 0.03), and lymphatic invasion (HR 2.34. *p* = 0.027) were associated with increased risk of mortality. In multivariate analyses, CAP grade (CAP 3 versus CAP 0–2, HR 2.44 (1.06–5.60), *p* = 0.035) was the only negative prognostic factor for survival (Table 4)**.**

### 3.6. Quality of Life

QoL questionnaires were sent to patients at set intervals. There was a substantial dropout of returned questionnaires after surgery (baseline n = 44, preoperative n = 43, 6 weeks postoperative n = 14, 3 months postoperative n = 29, 6 months postoperative n = 29, 12 months postoperative n = 10). The models for repeated measures of functioning scales demonstrated that there was no significant difference between BRPC and LAPC (*p* = 0.824) in terms of changes in global QoL over time (*p* = 0.064) (Appendix A). Combined for BRPC and LAPC patients, a significant positive change in global QoL was observed at 3 months postoperatively (*p* = 0.031) (Figure 3a). There was no significant difference in physical functioning (*p* = 0.483) over time (*p* = 0.271) (Figure 3a). Social and role functioning showed significant improvement 3 months postoperatively (*p* = 0.024, *p* = 0.031) (Figure 3a). There was no significant difference in emotional or cognitive functioning or financial impact over time (Figure 3a). There was significant improvement over time in pain (*p* = 0.042), dyspnea (*p* = 0.004), appetite loss (*p* = 0.028), and diarrhea (*p* = 0.007) from time of surgery (Figure 3b). There were no significant differences in the other symptom scales over time.

## 4. Discussion

This population-based study demonstrates that surgical resection for BRPC and LAPC following neoadjuvant chemotherapy was performed with a postoperative major morbidity rate of 25.9%, no postoperative 90-day mortality, and a median OS of 31 months. These findings suggest that patient selection was appropriate. Additionally, the cohort reported recovery of QoL across both functioning and symptom scores from 3 months post-surgery. Pathologic assessment showed an 87% R1 rate and a 79.7% N1/N2 lymph node involvement rate. No preoperative predictors of OS were identified. The only independent predictor of OS was the grade of tumor regression.

To date, few studies have reported on QoL in patients after resection of BRPC and LAPC. A recent Dutch multicenter study reported QoL in 29 patients undergoing chemotherapy and subsequent resection [25]. However, in line with this study, most studies experience a relatively high dropout rate due to disease progression or toxicity [25]. Addressing QoL in patients with pancreatic cancer is important given the short life expectancy. Furthermore, surgery in BRPC and LAPC requires patients to endure a clinical pathway that includes a lengthy and rigorous course of chemotherapy followed by complex surgery. Pancreatic resection is associated with a short-term deterioration in QoL, which usually returns to baseline values after 3–6 months [26]. In this study, almost half of the patients experienced grade 3–4 adverse events during neoadjuvant chemotherapy, and a quarter of the patients experienced major postoperative complications. However, a significant improvement in global QoL, social, and role functioning was demonstrated from 3 months postoperatively. Additionally, symptom scoring revealed a significant improvement in pain, dyspnea, appetite loss, and diarrhea from 3 months postoperatively.

The median OS of 31 months is comparable with that reported by other studies on resected BRPC and LAPC [27,28,29,30]. There was no significant difference in median OS between BRPC and LAPC. In accordance with the recently published complete NORPACT-2 cohort of a total of 188 resected and unresected patients, surgical resection improved survival outcomes compared to continued palliative chemotherapy in both BRPC and LAPC [5]. Furthermore, OS in the resected patients is comparable to that found in patients with primary resectable pancreatic cancer [27,28]. Non-therapeutic laparotomy may impact the chance of receiving further systemic chemotherapy and OS. The 13 patients undergoing a non-therapeutic laparotomy had a median OS of 13.1 months, and 61.5% continued with palliative chemotherapy. Although small numbers, these findings are in line with a transatlantic multicenter study of 663 patients receiving induction (m)FOLFIRINOX [29]. In that study, 67 (28.2%) patients underwent a non-therapeutic laparotomy with a median OS of 23.6 months, and 73.1% received palliative therapy after surgical exploration [29].

To enhance patient stratification and surgical decision-making in BRPC and LAPC, various predictors for OS have been identified. However, no preoperative predictors useful for patient selection were identified in the current study, in line with other studies [30]. This is likely due to the fact that the decision by the multidisciplinary team to recommend surgical exploration was individualized based on the degree of radiological and biochemical response, comorbidity profile, and performance status. Baseline CA19-9 > 500 mmol/L and receiving primary gemcitabine monotherapy were significantly associated with poor survival in the 188 patients in the NORPACT-2 cohort initiating primary chemotherapy [5]. However, in resected patients, neither CA19-9 levels at baseline or restaging, both as absolute values and categorized by response, nor the primary chemotherapy regimen were identified as independent predictors for OS.

In the current study, age > 70, CA19-9 > 150 at response evaluation, N2 status, poor tumor regression, and lymphatic invasion were associated with poor survival in univariate analysis. Only CAP grade remained significant in multivariate analyses. The CAP groups were dichotomized to evaluate tumor regression (CAP 3 versus CAP 0–2). CAP 3, i.e., extensive residual cancer with no evident tumor regression, was associated with poor survival (HR 2.44, *p* = 0.035). There is considerable divergence in opinion between pathologists when it comes to the identification of tumor regression [31]. 1. As the entire cohort was evaluated by a single, experienced pancreatic pathologist, interobserver variation was not an issue. Our data show that when carefully applied, the CAP tumor grading system may indeed provide prognostic stratification. Of studies reporting on the CAP system, two studies, including a total of 468 patients, have demonstrated a survival difference between grades (CAP 0–1 vs. CAP 2–3 and CAP 0 vs. CAP 3/CAP 1–2 vs. CAP3, respectively) [32,33,34].

The prognostic impact of resection margin status in resected pancreatic cancer following neoadjuvant treatment remains controversial. A recent meta-analysis demonstrated that R0 resections were significantly associated with prolonged overall and disease-free survival [35]. However, the significance diminished in the sensitivity analysis, including only partial pancreatoduodenectomies [35]. Differences in pre-treatment resectability status and variations in neoadjuvant treatment protocols (chemotherapy versus radiotherapy) may partially explain the inconsistencies among study results. In the current study, we found a high R1 rate of 85.2% based on <1 mm clearance and meticulous examination of the specimen surfaces. As discussed in the literature over the past decade, the R1 rate is highly dependent on the pathology examination method [36,37]. Especially, the use of axial specimen slicing has been shown to allow a more detailed evaluation of the pancreatic surfaces and, consequently, a higher detection rate of microscopic margin involvement than when other specimen dissection methods are used [36,38]. Furthermore, the extent of tissue sampling obviously has an impact on the detection of microscopic (i.e., macroscopically invisible) margin involvement [36]. Finally, using 1 mm clearance to define R1 (except for the anterior pancreatic surface), as recommended by (inter-)national guidelines, may also result in a higher R1 rate than when 0 mm clearance is considered as R1 [22]. The standardized pathology examination approach that was used for the study cohort combined both axial slicing, near-complete tissue embedding, and 1 mm clearance as the definition of R1, which explains the high R1 rate, as reported previously by our group and others [36,39,40,41]. The SMV margin was the most frequently affected margin (49%), followed by the SMA margin (37.3%). These findings are in accordance with our recent study that compared upfront pancreatectomy with or without venous resection and found that the R1 rate at the SMV groove was significantly higher in specimens with venous resection [92%] than in those without venous resection (50.7%; overall R1 rate: 82.7%) [41]. Involvement of multiple margins was also common (40.7%). The high R1 rate observed in this study indicates that the traditional aim of achieving an R0 resection in most cases is unrealistic in patients with BRPC and LAPC, of whom 59.3% underwent vascular resection. The high R1 rate in this study indicates that NAT does not have a major impact on the margins; the R1 rate is nearly identical to the one for treatment-naïve PDAC [41]. In our study, locoregional recurrence only and combined locoregional and distant recurrence rates were 28.2% and 23.1%, respectively. These rates are consistent with findings reported in the literature. For example, Stoop et al. found locoregional and combined recurrence rates of 14.3% and 35.3%, respectively, in a nationwide Dutch cohort [42]. Similarly, Groot et al. reported rates of 27.5% and 30.2%, respectively [43]. These data collectively underscore the ongoing challenge of locoregional recurrence in this patient population and highlight the need for improved strategies to reduce relapse risk.

The findings of this study should be interpreted in light of several limitations. First, this is a single-center study with a small study sample; thus, external validity may be challenged. Furthermore, the limited number of cases may have reduced the statistical power to detect significant differences. Second, QoL questionnaires sent to the patients showed a substantial dropout of patients returning the questionnaires after surgery. Patients struggling with recovery after complex surgery perhaps do not have the energy or mental surplus to complete the questionnaire, which may be a possible explanation. This attrition may introduce selection bias, as those with poorer postoperative outcomes could be underrepresented in the follow-up assessments, potentially resulting in overestimation of postoperative QoL improvements. The missing data also reduce the reliability and generalizability of our findings, highlighting the need for cautious interpretation of the results and for future studies employing strategies to minimize attrition. Third, although we found significant improvement of global QoL, social and role functioning, and symptoms of pain, appetite loss, dyspnea, and diarrhea over time, the absence of longitudinal univariate and multivariate analyses associating changes in QoL with clinical outcomes, such as survival, limits our ability to draw conclusions about the prognostic significance of these patient-reported outcomes. Future studies with appropriate designs and larger sample sizes are needed to investigate these potentially valuable associations. This study was a population-based study with a catchment area of 3.1 million people; hence, selection bias is limited. The study was carried out at a high-volume center with vast experience in both hepato–pancreato–biliary and transplant surgery, securing expertise and familiarity with both arterial and venous resections and reconstructions with potential challenging vascular anastomoses.

## 5. Conclusions

In conclusion, this population-based study found that resection in patients with BRPC and LAPC after receiving neoadjuvant chemotherapy was associated with a median OS of 31 months, no postoperative mortality, and a recovery of QoL from 3 months postoperatively. No preoperative predictors of OS were identified; however, poor histopathological tumor regression was significantly associated with worse survival. In patients with BRPC or LAPC who are considered for neoadjuvant chemotherapy followed by complex surgical procedures, it is essential that they receive thorough counseling at the time of diagnosis. The potential risks and benefits of both neoadjuvant chemotherapy and extensive surgery should be clearly communicated. Patients need to be informed about the possible adverse effects and complications, as well as the potential for improved survival and QoL. Future research should focus on identifying preoperative prognostic biomarkers for the selection of patients who will benefit from a surgical resection.

## Figures and Tables

**Figure 1 cancers-17-02505-f001:**
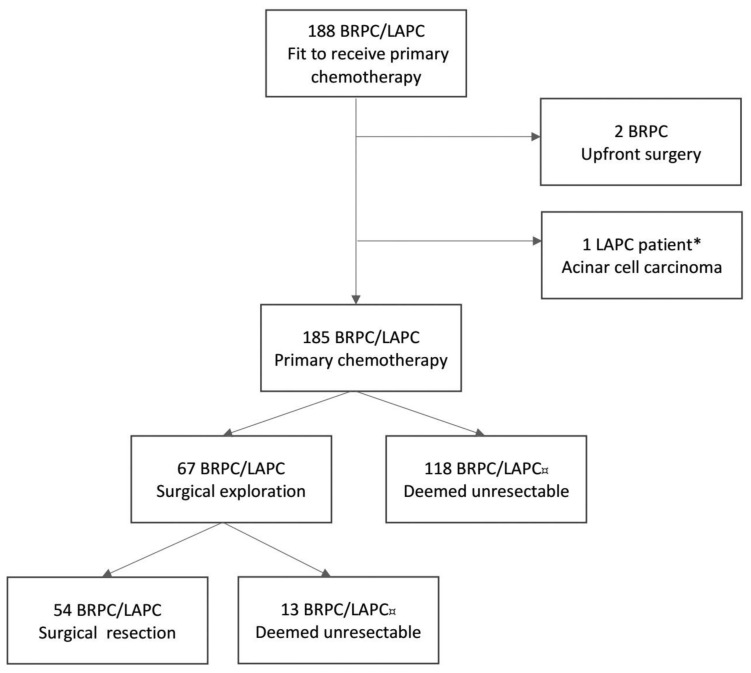
Flowchart of the NORPACT-2 study cohort [5]. * The patient underwent neoadjuvant chemotherapy and surgical resection. ¤ Patients deemed unresectable at response evaluations or during surgical exploration due to metastatic or locally advanced disease. BRPC, borderline resectable pancreatic cancer; LAPC, locally advanced pancreatic cancer.

**Figure 2 cancers-17-02505-f002:**
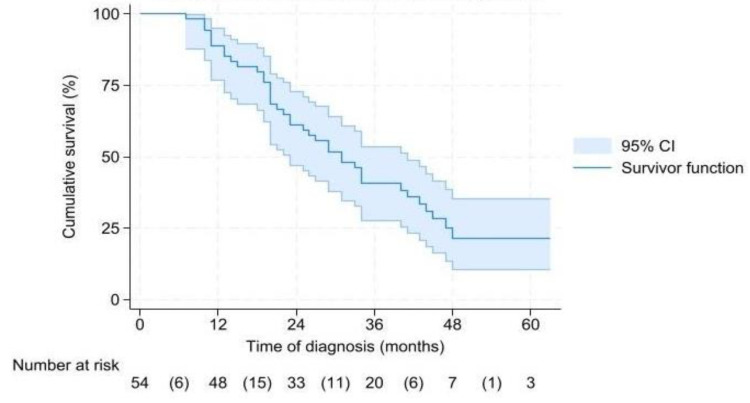
Overall survival from time of diagnosis.

**Figure 3 cancers-17-02505-f003:**
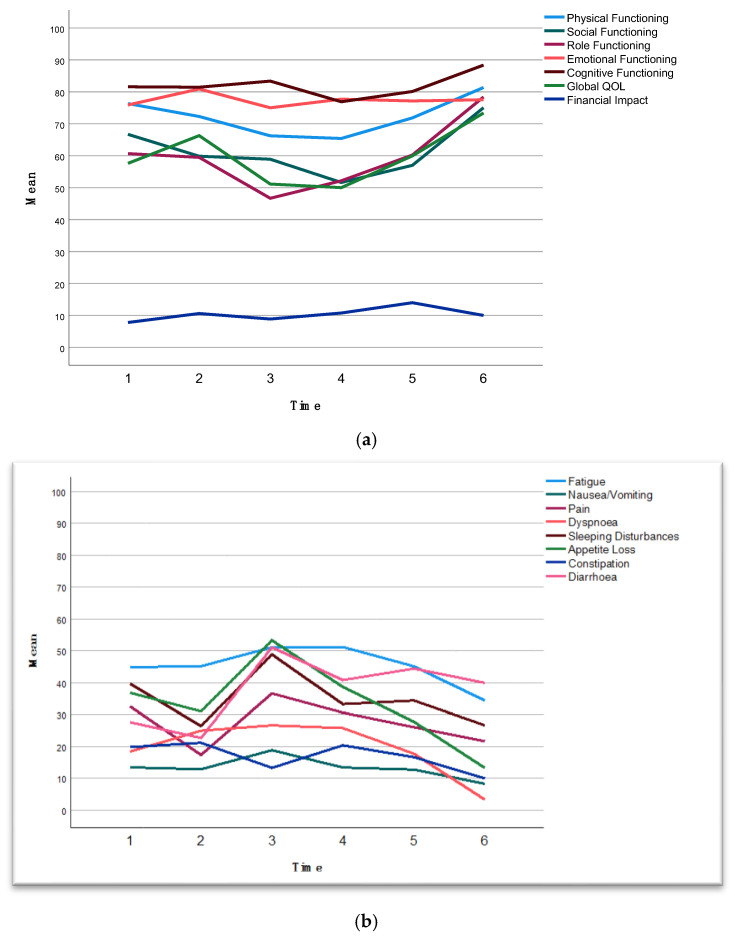
(**a**) The observed means at given time points for functioning scores of EORTC QLQ C-30. Higher scores on the functional scales indicate better functioning. For financial difficulties, a higher score indicates a greater level of financial difficulty. See Appendix A for estimated means with 95% confidence intervals. (**b**) The observed means at given time points for symptom scores of EORTC QLQ C-30. Higher scores on the symptom scales indicate more severe symptoms. See Appendix A for estimated means with 95% confidence intervals. Time points: 1, baseline; 2, before surgery after completion of neoadjuvant chemotherapy; 3, 6 weeks postoperatively; 4, 3 months postoperatively; 5, 6 months postoperatively; 6, 12 months postoperatively.

**Table 1 cancers-17-02505-t001:** Baseline and treatment characteristics of 54 patients with BRPC and LAPC undergoing neoadjuvant chemotherapy and surgical resection.

Characteristics	Value
Age, years, median (IQR)	68 (59–73)
Gender, n (%): male/female	31 (57.4)/23 (42.6)
Body mass index, median (IQR)	23.2 (20.2–27.2)
Charlson comorbidity index, n (%): 0/1/>1	29 (53.7)/17 (31.5)/9 (16.7)
Performance status—Eastern Cooperative Oncology Group, n (%): 0/1/2	33 (61.1)/19 (35.2)/2 (3.7)
Pre-treatment biliary drainage, n (%)	32 (59.3)
CA19-9 at diagnosis, kU/L, median (IQR)	191
CA 19-9 before surgery, kU/L, median (IQR)	97
CA19-9 normalization during neoadjuvant chemotherapy, n (%)	10 (18.5)
CA19-9 decrease > 50% during neoadjuvant chemotherapy, n (%)	22 (40.7)
Tumor location, n (%): head/body and tail	47 (87)/7(13)
Tumor diameter, mm, median (IQR)	30 (25–37)
Tumor classification, n (%): borderline resectable/locally advanced	43 (79.6)/11 (20.4)
Vascular involvement *, n (%): Portal–superior mesenteric vein	45 (83.3)
Superior mesenteric artery < 180	12 (22.2)
Superior mesenteric artery > 180	3 (5.6)
Hepatic artery	13 (24.1)
Celiac axis < 180	1 (1.9)
Celiac axis > 180	5 (9.3)
Neoadjuvant chemotherapy, n (%): (m)FOLFIRINOX	36 (66.7)
Gemcitabine-nab-paclitaxel	13 (24)
Gemcitabine	4 (7.4)
FLOX	1 (1.9)
Chemotherapeutic switch ¤	3 (5.6)
Number of neoadjuvant cycles, median (IQR)	4 (2–6)
Grade 3 or 4 adverse event during neoadjuvant chemotherapy, n (%)	26 (48.1)
RECIST response at restaging, n (%): Partial response	8 (14.8)
Stable disease	45 (83.3)
Progressive disease	1 (1.9)
Adjuvant chemotherapy, n (%):	33 (61)
mFOLFIRINOX	10 (18.5)
Gemcitabine/capecitabine	13 (24.1)
Gemcitabine	7 (13)
Gemcitabine-nab-paclitaxel	1 (1.8)
Gemcitabine/cisplatin	1 (1.8)
FLOX	1 (1.8)
Grade 3 or 4 adverse events during adjuvant chemotherapy, n (%)	9 (27.3)

Values are expressed as n (%) and median (IQR). * Percentages add to more than 100% since some patients presented with more than one vascular involvement. ¤ Three patients underwent chemotherapeutic switch from FOLFIRINOX to gemcitabine-nab-paclitaxel due to toxicity grade 3 and 4. CA19-9, carbohydrate antigen 19-9; (m)FOLFIRINOX, (modified) 5-fluorouracil with leucovorin, irinotecan, and oxaliplatin; FLOX, 5-fluorouracil with oxaliplatin; RECIST, Response Evaluation Criteria in Solid Tumors.

**Table 2 cancers-17-02505-t002:** Surgical outcomes.

Operation Type, n (%)	
Pylorus-preserving/classical pancreatoduodenectomy	21 (38.9)/25 (46.3)
Distal pancreatectomy	3 (5.6)
Total pancreatectomy	5 (9.3)
Vascular resection, n (%)	32 (59.3)
Venous resection	26 (48.1)
Combined venous and arterial resection	6 (11.1)
Periarterial divestment	1 (1.9)
Operative time, min, median (IQR) *	444 (361–558)
Estimated blood loss, mL, median (IQR) ¤	400 (225–750)
Length of stay, days, median (IQR)	7 (5–11)
90-day postoperative complications, n (%):	
All grades	40 (74.1)
Clavien grade ≥ 3	14 (25.9)
Delayed gastric emptying (A/B/C)	8 (14.8) (4/2/2)
Pancreatic-specific complications	3 (5.6)
Biochemical leak	2
Pancreatic fistula B/C	1
Post-pancreatectomy hemorrhage Reoperation ^#^	1 (1.8) 3 (5.6)
90-day postoperative mortality	0

Values are expressed as n (%) and median (IQR). * One patient with missing data. ¤ Three patients with missing data. ^#^ Causes included wound dehiscence/hemorrhage, liver necrosis, and portal vein thrombosis.

**Table 3 cancers-17-02505-t003:** Histopathological outcomes.

pT status pT0	1 (1.9)
pT1	4 (7.4)
pT2	34 (63)
pT3	14 (25.9)
pT4	1 (1.9)
pN status pN0	11 (20.4)
pN1	25 (46.3)
pN2	18 (33.3)
Tumor size, mm, median (IQR)	33 (27–39)
Lymph nodes retrieved, median (IQR)	21 (17–29)
Lymph node ratio, median (IQR)	0.08 (0.05–0.18)
Microvascular invasion: yes/no	21 (38.9)/33 (61.1)
Perineural invasion: yes/no	50 (92.6)/4 (7.4)
Lymphatic invasion: yes/no	37 (68.5)/17 (31.5)
Tumor regression grade (CAP) *: 0	1(1.9)
1	3 (5.7)
2	26 (48.1)
3	23 (42.6)
Margin status R0	7 (13)
R1	46 (85.2)
R2	1 (1.9)

Values are expressed as n (%) and median (IQR). ***** One patient underwent surgery abroad. Tumor regression grade was missing in the pathology report. CAP, College of American Pathologists; pT, tumor status; pN, lymph node status. ¤ R2 based on the pathology report with one grossly positive resection margin.

**Table 4 cancers-17-02505-t004:** Uni-and multivariate analysis of variables associated with overall survival (Cox regression) in 54 patients.

		Univariable		Multivariable	
	Number	HR (95% CI)	*p*-Value	HR (95% CI)	*p*-Value
Gender: Female	23 (42.6)	0.74 (0.39–1.41)	0.360		
Male (ref)	31 (57.4)				
Age (years): ≤70 (ref)	31 (57.4)				
>70	23 (42.6)	2.29 (1.21–4.33)	0.011	1.55 (0.68–3.49)	0.290
Performance status (ECOG): 0 (ref)	33 (61.1)				
1–2	21 (38.9)	1.53 (0.81–2.89)	0.192		
Body mass index: <18 (ref)	44 (81.5)				
18.5–30	5 (9.3)	0.97 (0.29–3.19)	0.962		
≥30	5 (9.3)	0.67 (0.13–3.35)	0.628		
Charlson comorbidity index: 0 (ref)	29 (53.7)				
1	16 (29.6)	0.82 (0.39–1.71)	0.596		
>1	9 (16.7)	0.98 (0.41–2.32)	0.963		
Tumor size (mm): 0–30 (ref)	29 (53.7)				
>30	25 (46.3)	0.76 (0.40–1.43)	0.390		
Location: Head/neck/uncinate process	47 (87)	1.69 (0.60–4.76)	0.322		
Body/tail (ref)	7 (13)				
Tumor classification: Borderline resectable (ref)	43 (79.6)				
Locally advanced	11 (20.4)	0.88 (0.41–1.93)	0.760		
CA19-9 at time of diagnosis: <37 (ref)	8 (14.8)				
≥37 to <150	11 (20.4)	0.52 (0.15–1.82)	0.310		
>150	25 (46.3)	1.60 (0.60–4.27)	0.347		
Primary chemotherapy: FOLFIRINOX or FLOX (ref)	37 (68.5)				
Gem/Nab-paclitaxel or Gem	17 (31.5)	2.18 (1.15–4.13)	0.017	0.85 (0.30–2.40)	0.765
CA19-9 at time of restaging: <37 (ref)	17 (31.5)				
≥37 to <150	15 (27.8)	1.02 (0.41–2.55)	0.958	1.02 (0.28–2.17)	0.624
>150	19 (35.2)	2.61 (1.19–5.75)	0.017	1.45 (0.53–3.9)	0.460
CA19-9 normalization: Yes	10 (18.5)	0.42 (0.16–1.08)	0.071		
No (ref)	41 (75.9)				
CA19-9 decrease > 50%: Yes	22 (40.7)	0.93 (0.46–1.88)	0.831		
No (ref)	22 (40.7)				
RECIST response: Complete/partial (ref)	8 (14.8)				
Stable/Progressive disease	46 (85.2)	1.11 (0.49–2.54)	0.792		
Type of surgery: Pancreatoduodenectomy (ref)	46 (85.2)				
Distal pancreatectomy	3 (5.6)	0.25 (0.34–1.84)	0.174		
Total pancreatectomy	5 (9.3)	1.05 (0.37–2.97)	0.925		
Portomesenteric vascular resection: No (ref)	22 (40.7)				
Yes	32 (59.3)	0.91 (0.48–1.73)	0.777		
Major morbidity (≥Clavien grade 3): No (ref)	40 (74.1)				
Yes	14 (25.9)	1.33 (0.67–2.64)	0.420		
Margin status: R0 (ref)	7 (13)				
R1/R2	47 (87)	1.56 (0.553–4.42)	0.399		
Tumor status: T0–T2 (ref)	39 (72.2)				
T3–T4	15 (27.8)	1.45 (0.73–2.87)	0.291		
Lymph node status: N0 (ref)	11 (20.4)				
N1	25 (46.3)	2.13 (0.79–5.74)	0.136		
N2	18 (33.3)	3.06 (1.11–8.36)	0.030		
Lymph node ratio		6.07 (0.97–37.9)	0.054		
Microvascular invasion: No (ref)	33 (61.1)				
Yes	21 (38.9)	1.3 (0.67–2.40)	0.475		
Perineural invasion: No (ref)	4 (7.4)				
Yes	50 (92.6)	4.45 (0.61–32.5)	0.141		
Lymphatic invasion: No (ref)	17 (31.5)				
Yes	37 (68.5)	2.34 (1.10–4.98)	0.027	1.79 (0.73–4.36)	0.201
Tumor regression grade (CAP) *: 0–2 (ref)	30 (55.6)				
3	23 (42.6)	3.36 (1.73–6.53)	<0.001	2.44 (1.06–5.60)	0.035
Pathology-based tumor size (mm): 0–30 (ref)	23 (42.6)				
>30	31 (59.3)	1.03 (0.99–1.05)	0.126		

Values are n (%) or hazard ratio (95% confidence intervals). BMI, body mass index; CA19-9, carbohydrate antigen 19-9; CAP, College of American Pathologists; ECOG, Eastern Cooperative Oncology Group; FLOX, 5-fluorouracil with oxaliplatin; FOLFIRINOX, 5-fluorouracil with leucovorin, irinotecan, and oxaliplatin; RECIST, Response Evaluation Criteria in Solid Tumors. * One patient underwent surgery abroad, and the CAP grade was not documented in the pathology report.

## Data Availability

The data presented in this study are available upon reasonable request from the corresponding author. The data are not publicly available due to privacy regulations.

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
