# Peer review of "Surgical, Histopathological, and Quality of Life Outcomes Following Neoadjuvant Chemotherapy and Pancreatectomy for Borderline Resectable and Locally Advanced Pancreatic Cancerâ€"

_cancers, 2025, doi:10.3390/cancers17152505_

Round 1
Reviewer 1 Report
Comments and Suggestions for Authors
Summary
This study follows 54 patients with borderline resectable (n=43) or locally advanced (n=11) pancreatic cancer who received neoadjuvant chemotherapy and then underwent pancreatectomy in a Norwegian health region. The authors report a median overall survival of 31 months, a 25.9 % rate of major (Clavien–Dindo ≥ III) complications, and an 85.2 % R1 resection rate. They also show that certain quality-of-life measures improve by three months after surgery. In their multivariate analysis, only a high CAP tumor regression grade (CAP 3) was linked to worse survival.
Strengths
The population-based design captures all eligible patients in the region, reducing referral bias and reflecting routine clinical practice.
Combining surgical details, centralized pathology, and longitudinal quality-of-life data provides a more complete picture of patient outcomes.
Having a single pathologist score margins and regression grades helps ensure consistency.
Recommendations
A flow chart tracking patients from diagnosis through chemotherapy, surgical exploration, and final resection would clarify how many were excluded at each step and why.
It would help to see a comparison of baseline characteristics between patients who ultimately had resection and those who did not, so we can understand potential selection bias.
Please explain exactly how the pathology team processed specimens—how were margins inked and measured? That would help readers interpret why 85.2 % of cases were R1.
Given the high R1 rate, comparing this method to those used at other centers would make it clearer whether this reflects tumor biology or different pathology techniques.
It’s unclear how variables were chosen for the Cox model. Clarify whether they used a univariate p < 0.10 cutoff or selected covariates beforehand, and report all variables tested.
Please state whether you checked proportional hazards assumptions (e.g., with Schoenfeld residuals); if not, consider commenting on potential violations.
Quality-of-life response rates dropped sharply (44 of 54 at baseline, 14 at six weeks, 29 at three months). Describe how you handled missing data—did you use mixed-effects models or imputation? Without this, improvements might overestimate recovery.
With only eleven LAPC patients, comparing BRPC vs. LAPC quality-of-life changes may not be reliable. It might be better to focus on overall trends rather than subgroup differences.
Statements comparing survival to primary resectable cases need context—either include a matched control or reference specific published benchmarks instead of implying a direct comparison.
Ensure figures show confidence intervals or error bars and clearly label scales (0–100), explaining that higher scores mean better function or worse symptoms as appropriate.
correct “27–399 mm” to “27–39 mm” for tumor size IQR and standardize percentage formatting (e.g., “25.9 %”).
Specify whether radiologic restaging (RECIST) was centrally reviewed or done locally.
Break down major complications by type (e.g., pancreatic fistula, hemorrhage, infection) to give readers a clearer sense of postoperative risks.
Define all abbreviations—SMA, ECOG, CA19-9—when they first appear.
Conclusion
This real-world cohort offers valuable insight into neoadjuvant therapy followed by resection for borderline and locally advanced pancreatic cancer. To improve clarity and strengthen the conclusions, the authors should add a detailed patient flow chart, explain how they managed missing quality-of-life data, clarify margin assessment methods, and more carefully compare survival results to established benchmarks.
Author Response
This study follows 54 patients with borderline resectable (n=43) or locally advanced (n=11) pancreatic cancer who received neoadjuvant chemotherapy and then underwent pancreatectomy in a Norwegian health region. The authors report a median overall survival of 31 months, a 25.9 % rate of major (Clavien–Dindo ≥ III) complications, and an 85.2 % R1 resection rate. They also show that certain quality-of-life measures improve by three months after surgery. In their multivariate analysis, only a high CAP tumor regression grade (CAP 3) was linked to worse survival.
Strengths: The population-based design captures all eligible patients in the region, reducing referral bias and reflecting routine clinical practice. Combining surgical details, centralized pathology, and longitudinal quality-of-life data provides a more complete picture of patient outcomes. Having a single pathologist score margins and regression grades helps ensure consistency.
Recommendations
Comment 1: A flow chart tracking patients from diagnosis through chemotherapy, surgical exploration, and final resection would clarify how many were excluded at each step and why.
Response: A flowchart has been added as Figure 1 on page 4.
Comment 2: It would help to see a comparison of baseline characteristics between patients who ultimately had resection and those who did not, so we can understand potential selection bias.
Response: Univariate and multivariate analysis of variables associated with surgical resection (logistic regression) and overall survival (Cox regression) in patients fit to receive chemotherapy using baseline factors for all patients in the NORPACT-2 cohort were published in BJS Open 2023 (Reference 5). It was out of the scope of this paper to redo these analyses. The following is added to the Introduction on page 2, line 64-66: “In the multivariate analysis of baseline factors associated with resection, LAPC , CA19-9 >500, and treatment with gemcitabine were negative predictors of eventual resection in 188 patients initiating primary chemotherapy in that study (5)”.
Comment 3: Please explain exactly how the pathology team processed specimens—how were margins inked and measured? That would help readers interpret why 85.2 % of cases were R1.
Response: Detailed information on the pathology examination procedure with updated references has been included in the Methods on page 3, line 114-133.
Comment 4: Given the high R1 rate, comparing this method to those used at other centers would make it clearer whether this reflects tumor biology or different pathology techniques.
Response: The impact of each step of the pathology examination process on the R1-rate is discussed, and relevant references are included in the extended Discussion on page 12, line 352-388.
Comment 5: It’s unclear how variables were chosen for the Cox model. Clarify whether they used a univariate p < 0.10 cutoff or selected covariates beforehand, and report all variables tested.
Response: Our sample was extensive given the type of cancer patients, however from a statistical point of view the sample size and consequently the statistical power was limited. In addition, all our analyses were exploratory, which is acknowledged in the Limitation section on page 13, line 389-392. However, we had a large selection of possible predictive factors which were all carefully collected and there was no missing data on these variables. We have analysed all available possible predictive variables in univariate Cox regression models and the results are reported in Table 4. Given the limited sample size and number of tested possible predictors, we used p<0.05 as a cut-off value for inclusion into a multivariable model. We have added to the Methods section on page 4, line 153-154: "Variables that were statistically significant in univariate analyses were included into a multivariable model".
Comment 6: Please state whether you checked proportional hazards assumptions (e.g., with Schoenfeld residuals); if not, consider commenting on potential violations.
Response: The assumption of proportional hazards is the key assumption for validity of the results derived using Cox model, therefore we have checked this for all univariate models and the final model using Schoenfeld residuals and visual inspection of plots. There were no violations to this assumptions. However, we are fully aware that the sample size and statistical power were limited which is reflected in low precision, e.g. wide confidence intervals of our estimates (please see Table 4 for details).
Comment 7: Quality-of-life response rates dropped sharply (44 of 54 at baseline, 14 at six weeks, 29 at three months). Describe how you handled missing data—did you use mixed-effects models or imputation? Without this, improvements might overestimate recovery.
Response: We are aware of possible selection bias and the high drop out rate. However, given the limited sample size we were not able to perform any model based imputations. Any other form of simple imputation methods would in our opinion only increase possible biases in our sample. Therefore, no imputation was performed, but we used generalized mixed models (GLM) for repeated measures so that we used all available data - GLM approach does not require complete data and thus imputation is not necessary.
Comment 8: With only eleven LAPC patients, comparing BRPC vs. LAPC quality-of-life changes may not be reliable. It might be better to focus on overall trends rather than subgroup differences.
Response: We agree that subgroup differences between BRPC and LAPC is not reliable. In fact we have not focused on subgroup differences except for one of the scores in EORTC-QLQ30: Global QOL. This figure has now been moved to the supplementary in the revised manuscript (Supplementary Figure S1).
Comment 9: Statements comparing survival to primary resectable cases need context—either include a matched control or reference specific published benchmarks instead of implying a direct comparison.
Response: In our manuscript, we have referenced two studies to provide this context. Specifically, we cite the work of Nymo et al. (reference no 27), which details survival outcomes in primary resectable pancreatic cancer within a Norwegian cohort, and van Dam et al. (reference no 28), which presents survival benchmarks from multiple international cohorts. These references are intended to serve as published benchmarks against which our findings are discussed.
Comment 10: Ensure figures show confidence intervals or error bars and clearly label scales (0–100), explaining that higher scores mean better function or worse symptoms as appropriate.
Response: We have now clearly labeled the axes in Figures 3a and 3b to indicate the scale (0–100). Additionally, the figure legends now clarify that higher scores on the functional scales indicate better functioning, while higher scores on the symptom scales indicate more severe symptoms. The figure comparing Global QoL between BRPC and LAPC has been moved to the supplementary (Supplementary Figure S3).
Regarding confidence intervals, we recognize their importance in representing data variability and have thus included this information in a supplementary table (Table S4), as including them directly on the figures would compromise clarity and readability due to the density of data points. This approach ensures all statistical information is available and accessible. We have referenced supplementary Table S4 both in the figure legend to Figure 3a and 3b. We hope this revision meets your expectations.
Comment 11: Correct “27–399 mm” to “27–39 mm” for tumor size IQR and standardize percentage formatting (e.g., “25.9 %”).
Response: Corrected and standardized throughout the manuscript.
Comment 12: Specify whether radiologic restaging (RECIST) was centrally reviewed or done locally.
Response: Added to the Methods section on page 3, line 98-99. “………., and reviewed at OUH by an experienced team of abdominal radiologists”.
Comment 13: Break down major complications by type (e.g., pancreatic fistula, hemorrhage, infection) to give readers a clearer sense of postoperative risks.
Response: Thank you for your valuable suggestion. We have revised the Results section to provide a detailed breakdown of major complications by Clavien-Dindo grade, type of complication, and corresponding intervention. For clarity and ease of reference, we have included this information in Supplementary Table S1. We believe this addition will give readers a clearer understanding of the spectrum and management of postoperative risks.
Comment 13: Define all abbreviations—SMA, ECOG, CA19-9—when they first appear.
Response: We have carefully reviewed the manuscript for all abbreviations, including SMV, SMA, and CA19-9, and have ensured that each is defined at first mention. Additionally, as per the journal’s guidelines, a comprehensive list of abbreviations has been provided in the manuscript.
Comment 14: This real-world cohort offers valuable insight into neoadjuvant therapy followed by resection for borderline and locally advanced pancreatic cancer. To improve clarity and strengthen the conclusions, the authors should add a detailed patient flow chart, explain how they managed missing quality-of-life data, clarify margin assessment methods, and more carefully compare survival results to established benchmarks.
Response: Thank you for your thoughtful and constructive feedback on our manuscript. We greatly appreciate the time and effort you dedicated to reviewing our work. Your insightful suggestions have significantly improved the quality of our paper. Thank you again for your valuable input.
Reviewer 2 Report
Comments and Suggestions for Authors
The work titled " Surgical, histopathological and quality of life outcomes following neoadjuvant chemotherapy and pancreatectomy for borderline resectable and locally advanced pancreatic cancer" represents a significant advancement in the field of cancer field.
In this study, the authors were aimed to investigate the impact of neoadjuvant chemotherapy followed by resection on overall survival in cancerous patients. The findings showed that more than fifty% of patients with BRPC; and the majority received folfirinox chemotherapy drug. More than Forty of the collected sample of patients underwent pancreatoduodenectomy. The patients with BRPC and LAPC undergoing neoadjuvant chemotherapy and resection have survival comparable to primary resectable pancreatic cancer. Postoperative morbidity was acceptable and QoL recovered post-surgery. CAP grade was the only inde-pendent negative prognostic factor.All of these findings seem valuable and suitable contribution to be published in the Cancers journal after justifying some points:
- One of the limitations of this study the sample collecting time is a little bit old 2018-2020, why the authors did not collect recent data rather than old data?
- It is recommended to update the title to be more attractive for the readers and researchers
- The number of patients the sample is one of the other limitations that should be discussed well because the sample size is small rationally
- In the abstract you mentioned that the “The majority 34(66.7%) received (m)FOLFIRINOX.” So what about the others what did they receive
- In the abstract you mentioned major complication it is recommended to mention some of them
- You mentioned in the abstract that “The majority of the resected specimens demonstrated T2 (63%), N+ (79.6%), and R1 (85.2%) status.” If this symbols or abbreviations clear for you maybe it is not for the readers so it is recommended to mention them
- The introduction is premature and should be updated with recent work, for example the statistical data were cropped from work published in 2014 but you should write the updated data since we are 2025
- The sample size calculation should be mentioned in the method section
- Did the authors make the validation test before start collecting the data
- In the results you write “Overall, 55 patients” while in the abstract you wrote 54 patients ??
- Regarding the Table 1 data each column should be started with head or title regarding the data below
- It seems that figure 1 and figure S1 are the same? if yes remove this figure from the supplementary file
- The conclusion should be improved more by adding limitation and ethical considerations
Best wishes
Author Response
Reviewer 2
The work titled " Surgical, histopathological and quality of life outcomes following neoadjuvant chemotherapy and pancreatectomy for borderline resectable and locally advanced pancreatic cancer" represents a significant advancement in the field of cancer field. In this study, the authors were aimed to investigate the impact of neoadjuvant chemotherapy followed by resection on overall survival in cancerous patients. The findings showed that more than fifty% of patients with BRPC; and the majority received folfirinox chemotherapy drug. More than Forty of the collected sample of patients underwent pancreatoduodenectomy. The patients with BRPC and LAPC undergoing neoadjuvant chemotherapy and resection have survival comparable to primary resectable pancreatic cancer. Postoperative morbidity was acceptable and QoL recovered post-surgery. CAP grade was the only independent negative prognostic factor. All of these findings seem valuable and suitable contribution to be published in the Cancers journal after justifying some points:
Comment 1: One of the limitations of this study the sample collecting time is a little bit old 2018-2020, why the authors did not collect recent data rather than old data?
Response: The data collection period (2018–2020) coincides with the introduction and widespread adoption of modern chemotherapy regimens, which was essential for accurately assessing their impact in a real-world, population-based cohort. Additionally, this was a prospective study conducted at a high-volume center, designed to provide robust data on resection rates and survival outcomes.
We agree that more recent data are valuable. In fact, based on the findings and groundwork of the current study, a new prospective study (NORPACT-3) was initiated in 2024 and is currently ongoing. We believe our data from 2018-2020 remain relevant and provide valuable baseline information.
Comment 2: It is recommended to update the title to be more attractive for the readers and researchers
Response: We appreciate the reviewer’s suggestion to update the title to make it more attractive for readers and researchers. However, we respectfully prefer to retain the current title, as it accurately reflects the comprehensive analysis performed in this single-center, population-based cohort. We believe that the existing title appropriately conveys the scope and focus of our study, ensuring clarity for potential readers and researchers interested in this specific area. Thank you for your understanding.
Comment 3: The number of patients the sample is one of the other limitations that should be discussed well because the sample size is small rationally
Response: We have addressed this in the limitation paragpraph in the Discussion on page 12, line 389-392: “First, this is a single centre study with a small study sample, thus external validity may be challenged. Furthermore, the limited number of cases may have reduced the statistical power to detect significant differences».
Comment 3: In the abstract you mentioned that the “The majority 34 (66.7%) received (m)FOLFIRINOX.” So what about the others what did they receive
Response: The other patients received Gemcitabine-nab-paclitaxel (n=13), Gemcitabine (n=4) , or FLOX (n=1). These data are shown in Table 1.
Comment 4: In the abstract you mentioned major complication it is recommended to mention some of them
Response: Thank you for your valuable suggestion. We have revised the Results section to provide a detailed breakdown of major complications by Clavien-Dindo grade, type of complication, and corresponding intervention. For clarity and ease of reference, we have included this information in Supplementary Table S1. We believe this addition will give readers a clearer understanding of the spectrum and management of postoperative risks.
Comment 5: You mentioned in the abstract that “The majority of the resected specimens demonstrated T2 (63%), N+ (79.6%), and R1 (85.2%) status.” If this symbols or abbreviations clear for you maybe it is not for the readers so it is recommended to mention them
Response: We believe that these terms are familiar to our target readers and are commonly used in the surgical literature. Therefore, we have not made any changes to the text regarding these abbreviations.
Comment 6: The introduction is premature and should be updated with recent work, for example the statistical data were cropped from work published in 2014 but you should write the updated data since we are 2025
Response: We would like to clarify that the statistical data referenced are not from 2014. Instead, they are based on a publication from 2023, which included patients enrolled between 2018 and 2020, with available follow-up data. The study cohort is presented in the Flowchart in the revised manuscript (Figure 1).
Comment 7: The sample size calculation should be mentioned in the method section
Response: We have included all available patients. However, we are aware that our sample and thus statistical power is limited and all our analyses should be considered exploratory and hypotheses generating. This is acknowledged in the Limitation section on page 12 , line 389-392.
Comment 8: Did the authors make the validation test before start collecting the data
Response: In the context of a prospective, observational, population based study conducted over a defined time period, the term "validation test" could potentially refer to several different concepts, depending on study design and context. As this study includes all consecutive patients treated at our center over a defined period, we did not perform a formal validation test prior to data collection. If further clarification or information on our data collection and quality assurance procedures is needed, we are happy to provide more details. Could you kindly clarify whether you are referring to validation of the data collection process, questionnaires, or to external validation of the findings in another cohort?
Comment 9: In the results you write “Overall, 55 patients” while in the abstract you wrote 54 patients ??
Response: Yes, overall 55 patients with pancreatic cancer were resected. However, as mentioned in the next sentence in the original manuscript, one patient with acinar cell carcinoma was excluded, and only patients with pancreatic ductal adenocarcinoma were included in the analysis. We have changed the number of patients in the results from 55 to 54 for clarity. The flowchart (Figure 1) shows the cohort from which the patients were selected.
Comment 10: Regarding the Table 1 data each column should be started with head or title regarding the data below
Response: Each column is now started with a title (Characteristic and Value).
Comment 10: It seems that figure 1 and figure S1 are the same? if yes remove this figure from the supplementary file
Response: Figure 1 and Figure S1 are not the same. Figure 1 (in the revised manuscript Figure 2) shows overall survival from time of diagnosis. Figure S1 (in the revised manuscript Figure S2) shows overall survival from time of surgery.
Comment 11: The conclusion should be improved more by adding limitation and ethical considerations
Response: We have extended the paragraph discussing the main limitations of our study on page 12-13, line 389-411. Regarding ethical considerations, as stated in the Methods section, this study was reviewed and approved by the Regional Ethical Committee, and informed consent was obtained from all participants prior to participation and data collection. Patient confidentiality and privacy were strictly maintained throughout the study, and all procedures were conducted in accordance with the principles of the Declaration of Helsinki. In addition, we have added the following statements to the conclusion on page 13, line 417-423: “In patients with BRPC or LAPC who are considered for neoadjuvant chemotherapy followed by complex surgical procedures, it is essential that they receive thorough counseling at time of diagnosis. The potential risks and benefits of both neoadjuvant chemotherapy and extensive surgery should be clearly communicated. Patients need to be informed about the possible adverse effects and complications, as well as the potential for improved survival and QoL.»
Thank you for your thorough and constructive review. Your feedback has improved our manuscript, and we appreciate your valuable input.
Reviewer 3 Report
Comments and Suggestions for Authors
I would like to thank the authors for their valuable articles.
In the study results; I believe that quality of life outcomes are more important than other outcomes. Because, as can be seen in Table 2 of the article https://www.nature.com/articles/s41571-023-00746-1/tables/2, the effectiveness of neoadjuvant treatment has been evaluated in many studies with high patient numbers, and the patient numbers are quite high compared to the current study.
This study has quality of life outcomes that are candidates to contribute to the literature.
Suggestions-criticisms
-The number of patients is too limited to generalize from the literature
-In a study evaluating the effectiveness of neoadjuvant therapy, it would be valuable to include not only patients who underwent surgery but also patients with BRPC and LAPC who were given neoadjuvant chemotherapy but could not be operated on (disease progression or any additional reason), so that we can see how many patients we can actually operate on with neoadjuvant therapy, and thus we will be able to determine which patient group benefits from neoadjuvant chemotherapy
-Why did you not state that the presence of N2 disease (Lymph node status), which was statistically significant in the univariate analysis, was not included in the multivariate analysis?
-I recommend focusing on quality of life outcomes in the discussion section.
Best regards
Author Response
Reviewer 3
I would like to thank the authors for their valuable articles. In the study results; I believe that quality of life outcomes are more important than other outcomes. Because, as can be seen in Table 2 of the article https://www.nature.com/articles/s41571-023-00746-1/tables/2, the effectiveness of neoadjuvant treatment has been evaluated in many studies with high patient numbers, and the patient numbers are quite high compared to the current study. This study has quality of life outcomes that are candidates to contribute to the literature.
Thank you for your positive and encouraging feedback.
Comment 1: The number of patients is too limited to generalize from the literature
Response: We have addressed this in the limitation paragpraph in the Discussion on page 12, line 389-392: “First, this is a single centre study with a small study sample, thus external validity may be challenged. Furthermore, the limited number of cases may have reduced the statistical power to detect significant differences".
Comment 2: In a study evaluating the effectiveness of neoadjuvant therapy, it would be valuable to include not only patients who underwent surgery but also patients with BRPC and LAPC who were given neoadjuvant chemotherapy but could not be operated on (disease progression or any additional reason), so that we can see how many patients we can actually operate on with neoadjuvant therapy, and thus we will be able to determine which patient group benefits from neoadjuvant chemotherapy.
Response: Indeed, we agree that it is important to evaluate not only the patients who proceed to surgery but also those who receive primary chemotherapy yet do not undergo surgical resection, in order to fully assess the effectiveness of neoadjuvant therapy and better identify the subgroups who benefit.
As we already have noted in the submitted manuscript, this information—regarding the entire cohort, including patients who were not resected due to disease progression or other reasons—has already been published [Reference 5, Farnes et al, BJS Open 2023). In that referenced publication, we reported the outcomes and reasons for non-resection in the full cohort of 188 patients initially considered for neoadjuvant therapy.
The current manuscript specifically focuses on analyzing the 54 patients who underwent resection following neoadjuvant therapy, with the intention of providing a more detailed investigation of survival, histopathological and quality of life outcomes within this subgroup. We have clarified this point in the revised manuscript (see flowchart Figure S1), and the reference to the previously published data on page 2, line 62-66 and line 77-79.
Comment 3: Why did you not state that the presence of N2 disease (Lymph node status), which was statistically significant in the univariate analysis, was not included in the multivariate analysis?
Response: Since there is a high degree of correlation between N2 status and lymphatic invasion, we included only lymphatic invasion in the multivariate analysis.
Comment 4: I recommend focusing on quality of life outcomes in the discussion section.
Response: We thank the reviewer for this insightful suggestion. We have moved the discussion of quality of life outcomes to the second paragraph of the Discussion section on page 11, line 299-313. In addition, we have expanded the discussion og the limitations of the the quality of life outcomes on page 12-13, line 392-406.
Reviewer 4 Report
Comments and Suggestions for Authors
- The plaquarism need to reduce below 15%
- Rewrite the introduction part with more details
- Remodify the structure of Table 1 and Table 2
- Include some figures
- The QoL analysis is weakened by high attrition rates. Please discuss the impact of this missing data on reliability and address potential bias.
- The lack of longitudinal multivariate analysis associating QoL changes with clinical outcomes is a missed opportunity. Consider adding or acknowledging this limitation.
- The reported R1 rate of 85.2% appears unusually high. The use of a <1 mm margin clearance criterion should be justified, considering international differences in margin assessment. Additionally, please elaborate on how this high R1 rate influences the interpretation of survival outcomes, particularly given that R1 status was not identified as a significant prognostic factor in your analysis.
- The CAP grading interpretation should be clarified with stronger histopathological context and discussion of inter-observer variability, which is acknowledged only briefly.
- More detailed statistical methods for QoL data should be described.
- The recurrence data are presented but not thoroughly discussed.
- Rewrite the discussion part with more details and use recent references
- The figures and supplementary materials are not properly mentioned within the text.
- Check grammatical errors throughout the manuscript
Need to be improved
Author Response
Reviewer 4
Comment 1: The plaquarism need to reduce below 15%
Response: Any possible plagiarism may be due to references in the Introduction and Methods sections to previously published work (references 5 and 12), or the references to the EORTC scoring manual (reference 24). Instead of citing these as "as previously described," we chose to repeat some of the text in our manuscript. Otherwise, we believe that all quotations from other work are correctly cited with appropriate references.
Comment 2: Rewrite the introduction part with more details
Response: We have revised the introduction to provide more background information. Specifically, we have incorporated additional references to ensure a more thorough contextualization of the study within the existing literature. Furthermore, we have clarified the context and characteristics of the study cohort to enhance the reader’s understanding of its relevance.
Comment 3: Remodify the structure of Table 1 and Table 2
Response: We have modified Table 1 so that each column now begins with a clear title ("Characteristic" and "Value"). Aside from this adjustment, we have retained the original structure, as we believe it best presents the data relevant to our study objectives. However, we are open to making further revisions if the reviewer has specific recommendations.
Comment 4: Include some figures
Response: The original manuscript contained four main figures and two supplementary figures to illustrate our findings. In response to your feedback, we have added an additional figure—a flowchart of the study cohort (Figure 1)—to further enhance the clarity and presentation of our study design and patient selection process.
Comment 5: The QoL analysis is weakened by high attrition rates. Please discuss the impact of this missing data on reliability and address potential bias.
Response: Thank you for highlighting the impact of attrition on the reliability and potential bias in QoL analysis. We have expanded our discussion on page 12-13, line 392-406 to more thoroughly address how missing data may affect our estimates and underscore the importance of interpretive caution, as well as the need for future research with enhanced follow-up protocols.
Comment 6: The lack of longitudinal multivariate analysis associating QoL changes with clinical outcomes is a missed opportunity. Consider adding or acknowledging this limitation.
Response: We agree that longitudinal multivariate analysis associating QoL changes with clinical outcomes would provide valuable insights. However, as this was beyond the scope and objectives of our current study, we have explicitly acknowledged this limitation on page 13, line 400-406 in the revised manuscript and highlighted the need for future research addressing this important question.
Comment 7: The reported R1 rate of 85.2% appears unusually high. The use of a <1 mm margin clearance criterion should be justified, considering international differences in margin assessment. Additionally, please elaborate on how this high R1 rate influences the interpretation of survival outcomes, particularly given that R1 status was not identified as a significant prognostic factor in your analysis.
Response: The reasons for the high R1-rate are discussed on page 12, line 358-382, and relevant references are included, also regarding the use of 1 mm clearance as the definition for R1. The impact of microscopic margin involvement following neoadjuvant therapy is not well understood. Even if neoadjuvant treatment has a limited effect in most patients, the already small number of residual cancer cells in the surgical bed in case of microscopic margin involvement (R1) may have become even lower after completed preoperative chemotherapy. Consequently, the time needed for this tiny cancer cell focus to grow and become clinically manifest to the point of determining survival may well be longer than the time required for distant recurrence to develop. Hence, the presence of R1 may not necessarily correlate with shorter survival. Obviously, as long as accurate quantification of the residual cancer burden in the surgical bed and the time-dependent growth kinetics of these remain unknown, we can only hypothesize about the impact of R1 on overall survival.
Comment 8: The CAP grading interpretation should be clarified with stronger histopathological context and discussion of inter-observer variability, which is acknowledged only briefly.
Response: The histopathological description of CAP grading has been included in the Methods on page 3, line 127-133. As the entire cohort was evaluated by a single, experienced pancreatic pathologist, interobserver variation was not an issue, commented on page 11, line 345-347. Our data show that when carefully applied, the CAP tumour grading system may indeed provide prognostic stratification.
Comment 9: More detailed statistical methods for QoL data should be described.
Response: We have added to the Methods on page 4, line 154-159: “We have Generalised mixed models (GLM) for repeated measures to assess possible changes over time for QoL variables. We used identity link function as the outcome was continuous. GLM do not require complete data so we could use all available data despite drop outs and missing data. Moreover, we used unstructured covariance matrix as not to impose any structure on our data.”
Comment 10: The recurrence data are presented but not thoroughly discussed.
Response: We would like to clarify that the recurrence data were addressed in the original version of the manuscript, where we found that the recurrence rates are consistent with findings reported in the literature, page 12, line 382-388. We recognize that further clarification would be beneficial regarding the reported R1 rate observed in our cohort. In response to your suggestion, we have now expanded the discussion regarding the R1 rate observed in our cohort, page 12, line 358-382.
Comment 11: Rewrite the discussion part with more details and use recent references
Response: We have expanded the discussion section to provide more detailed analysis and critical insights. We have also updated the references to include several relevant publications to better contextualize our findings within the current literature. We kindly ask the reviewers to refer to the revised discussion for these updates, as many of the other reviewers had related suggestions which have also been addressed in this revision. We hope these improvements meet your expectations.
Comment 12: The figures and supplementary materials are not properly mentioned within the text.
Response: We have carefully reviewed the manuscript and checked all references to figures and supplementary materials, ensuring they are now properly and clearly mentioned at the appropriate places in the text.
Comment 13: Check grammatical errors throughout the manuscript
Response: We have thoroughly read through the entire manuscript and made revisions to improve clarity and address any grammatical errors we identified. We trust that the revised version meets the required language standards.
Thank you for your constructive criticism and thorough review. Your input has been very valuable.
Reviewer 5 Report
Comments and Suggestions for Authors
This paper describes a single-center experience of pancreatic cancer surgery performed after neoadjuvant therapy. To be fair, this is largely a confirmatory paper with a limited novelty. Nevertheless, I am highly positive about this manuscript:
1) Pancreatic cancer surgery is among the most complicated surgical procedures, with a limited number of centers specializing on this intervention; therefore, proper reporting of relevant experience of each particular hospital is of importance.
2) This retrospective study is exceptionally well-performed and well-presented. All sections of the manuscript are highly informative. The Discussion is well-balanced and meaningful.
Author Response
Comment: This paper describes a single-center experience of pancreatic cancer surgery performed after neoadjuvant therapy. To be fair, this is largely a confirmatory paper with a limited novelty. Nevertheless, I am highly positive about this manuscript: 1) Pancreatic cancer surgery is among the most complicated surgical procedures, with a limited number of centers specializing on this intervention; therefore, proper reporting of relevant experience of each particular hospital is of importance. 2) This retrospective study is exceptionally well-performed and well-presented. All sections of the manuscript are highly informative. The Discussion is well-balanced and meaningful.
Response: Thank you very much for your positive and encouraging feedback.
Round 2
Reviewer 2 Report
Comments and Suggestions for Authors
the authors were improved their manuscript well
Reviewer 3 Report
Comments and Suggestions for Authors
I would like to thank the authors for their valuable articles. However, when the literature on neoadjuvant treatments in pancreatic cancer is examined, there are many original articles and meta-analysis-reviews. Therefore, although it is well written, I think its contribution to the literature in this sense is limited.
I also think that not enough emphasis has been placed on quality of life, which seems to be a positive aspect of the study.
Best regards
Reviewer 4 Report
Comments and Suggestions for Authors
Recommend for Acceptance